# Enantioselective synthesis of chiral quinohelicenes through sequential organocatalyzed Povarov reaction and oxidative aromatization

Chengwen Li[1], Ying-Bo Shao[1], Xi Gao[1], Zhiyuan Ren[1], Chenhao Guo[2], Meng Li[2] & Xin Li [1,3] ✉

Heterohelicenes are of increasing importance in the fields of materials science, molecular recognition, and asymmetric catalysis. However, enantioselective construction of these molecules, especially by organocatalytic methods, is challenging, and few methods are available. In this study, we synthesize enantioenriched 1-(3-indol)-quino[n]helicenes through chiral phosphoric acid-catalyzed Povarov reaction followed by oxidative aromatization. The method has a broad substrate scope and offers rapid access to an array of chiral quinohelicenes with enantioselectivities up to 99%. Additionally, the photochemical and electrochemical properties of selected quinohelicenes are explored.

Helicenes, which have rigid *ortho*-fused π-conjugated polycyclic aromatic structures, are important because of their inherent chirality, which arises from their helicity. Because of their distinctive electronic properties, they have been widely investigated for their potential use in the fields of materials science and molecular recognition[1–6]. They can also serve as chiral ligands and catalysts[1,7–10] and have even been used to increase the activity of catalysts for the oxygen evolution reaction[11]. The potential utility of these molecules has triggered extensive investigation of the synthesis of chiral helicenes with novel structures and functional groups. But, conventional acquisition of chiral helicenes rely mainly on the resolution of racemic helicenes by means of chiral resolution reagents or chiral HPLC separation, or asymmetric syntheses enabled by chiral auxiliaries or chiral substrates[12–17].

However, in sharp contrast with central chirality and axial chirality, which are well-studied, helical chirality, and the catalytic enantioselective synthesis of helicenes, remain under-explored (Fig. 1A). In 1999, Stará et al. realized the first enantioselective synthesis of helicenes, through [2 + 2 + 2] cycloaddition reactions of triple alkynes catalyzed by a chiral nickel complex[18]. Since then,

transition-metal-catalyzed enantioselective [2 + 2 + 2] cycloaddition has frequently been used for asymmetric construction of multiple chiral helicenes[19–29]. Another effective method for the synthesis of enantioenriched helicenes involves gold-catalyzed enantioselective intramolecular hydroarylation of alkynes[30–35]. Other transition-metal-catalyzed approaches have been reported[36–39], including vanadium-catalyzed oxidative coupling of polycyclic phenols[36] and rhodium-catalyzed enantioselective C-H activation/annulation of 1-aryl isoquinoline derivatives and alkynes[38]. Despite these efforts, challenges associated with relatively high catalyst loadings, harsh reaction conditions, and limited substrate scope remain unsolved.

The great majority of enantioselective syntheses of helicenes involve transition-metal catalysts, and examples using organocatalytic strategies are limited in number (Fig. 1A). In 2014, List et al. reported enantioselective synthesis of azahelicenes via asymmetric Fischer indole reactions catalyzed by chiral phosphoric acid[40]. Yan et al. accomplished highly enantioselective synthesis of helicenes via vinylidene *ortho*-quinone methide intermediates with catalysis by chiral bifunctional amides[41,42]. In 2020, Bonne and Rodriguez et al. described asymmetric synthesis of dioxa[6]helicenes by a Michael/*O*-alkylation

[1]State Key Laboratory of Elemento-Organic Chemistry, College of Chemistry, Nankai University, Tianjin 300071, China. [2]Beijing National Laboratory for Molecular Sciences, CAS Key Laboratory of Molecular Recognition and Function, Institute of Chemistry, Chinese Academy of Sciences, Beijing 100190, China. [3]Haihe Laboratory of Sustainable Chemical Transformations, Tianjin 300192, China. ✉e-mail: xin_li@nankai.edu.cn

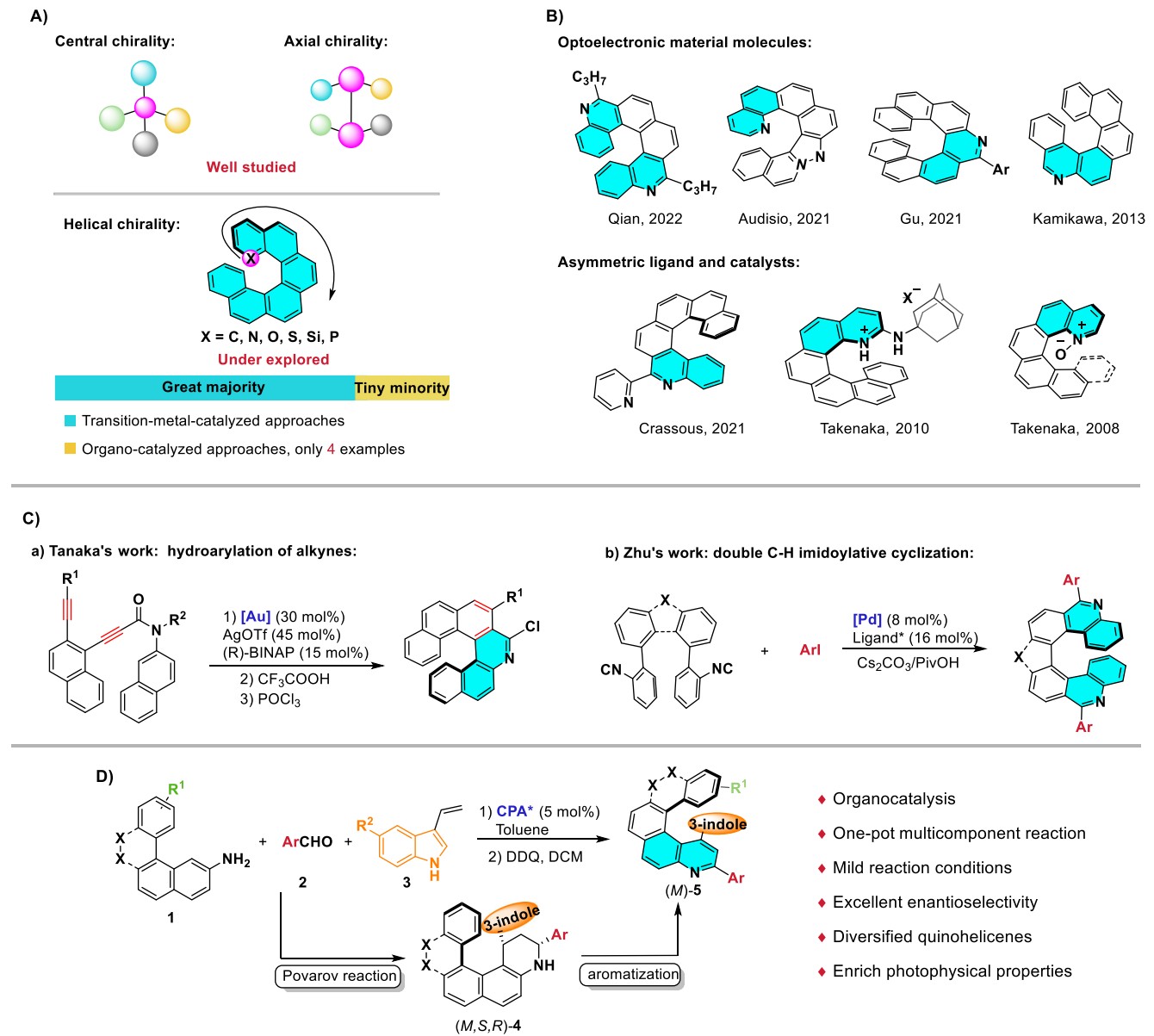

**Fig. 1 | Background introduction and overview of this work. A** The development of asymmetric synthesis of central-, axial- and helical- chirality. **B** Application examples of quino-containing helicenes. **C** Enantioselective synthesis of quinohelicenes. **D** Overview of this work.

heteroannulation process mediated by a bifunctional quinine-derived squaramide organocatalyst[43].

The increasing demand for enantioenriched helicene compounds in various fields has stimulated the development of efficient methods for stereoselective asymmetric synthesis of structurally diverse helicenes, but efficient methods involving organocatalysis would be highly desirable.

Of particular interest are quinohelicenes, a subclass of hetero-helicenes with potential applications in optoelectronics[9,44–51], asymmetric catalysis[7–9,52], and molecular recognition[53] (Fig. 1B). To date, only two transition-metal-catalyzed syntheses of chiral quinohelicenes have been reported. In 2014, Tanaka et al.[30] reported gold-catalyzed intramolecular hydroarylation of alkynes, but a high catalyst loading (30 mol%) was required, the substrate scope was limited (only two examples were reported), and the enantioselectivity was only moderate (74% ee) (Fig. 1C). Very recently, Zhu et al.[39] realized the synthesis of chiral quinohelicenes through palladium-catalyzed asymmetric double imidoylative cyclization (Fig. 1C). It should be noted that the molecules synthesized by these two methods had the quinoline heterocycle in the

middle of the polycyclic aromatic structure. There is an urgent need to develop a highly enantioselective method for the synthesis of quino-helicenes with a side quinoline ring.

One of the most attractive approaches for the construction of chiral quinoline-containing molecules is the Povarov reaction, which is an excellent method for building ring systems from simple, readily available substrates[54–62]. And, we found that the Povarov reaction catalyzed by lewis acid can be used to construct racemic quinohelicenes, even though which suffered from high catalyst loading and harsh conditions[63,64]. Inspired by these works, we speculated that combining a Povarov reaction catalyzed by a chiral Brønsted acid with a subsequent oxidative aromatization would allow the synthesis of chiral quinohelicenes. This approach not only would avoid the use of a transition metal but also would have high step-economy in that the chiral quinohelicene skeleton could be constructed in a one-pot reaction of three easily accessible raw materials. Herein, we report a combination strategy involving a chiral phosphoric acid-catalyzed Povarov reaction and subsequent oxidative aromatization with 1,2-dichloro-4,5-dicyanobenzoquinone (DDQ) to prepare a wide range of

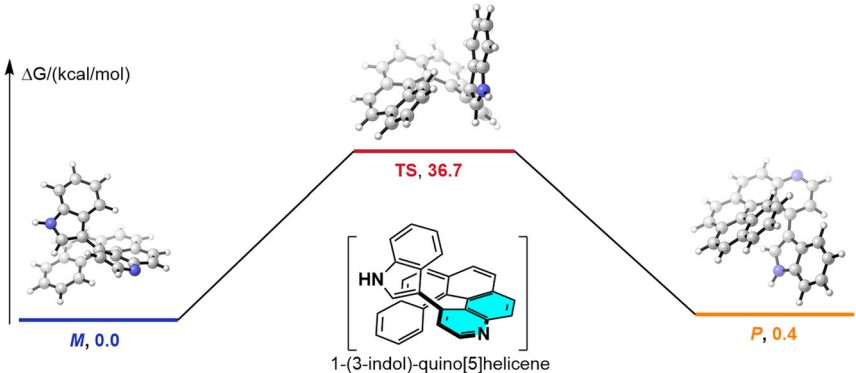

**Fig. 2 | Energy barrier to racemization of 1-(3-indol)-quino[5]helicene.** The relative Gibbs free energy (kcal/mol) was calculated at the SMD(Toluene)/M06-2X-D3/def2-TZVPP//B3LYP D3(BJ)/def -2-SVP level of theory.

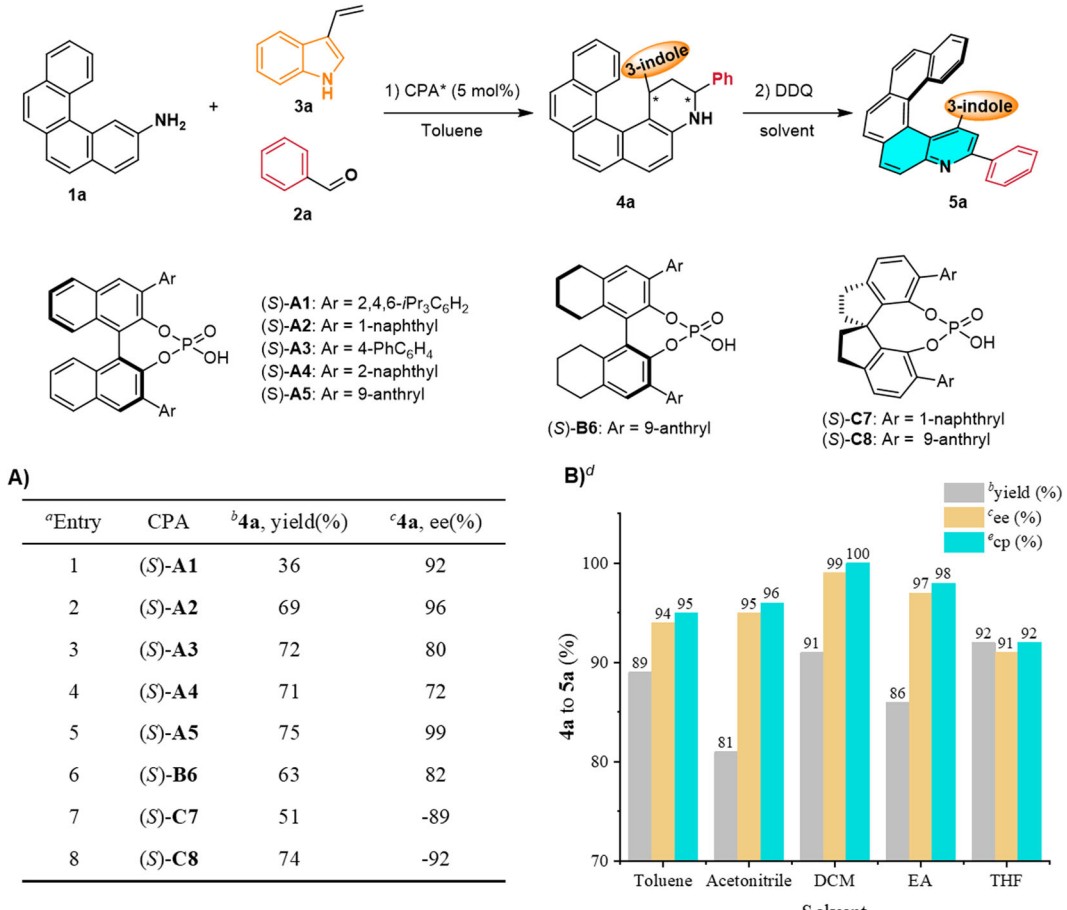

**Fig. 3 | Optimization of the reaction conditions.** Optimization of conditions for (**A**) the Povarov reaction and (**B**) the oxidative aromatization: [a]Povarov reaction conditions: **1a** (0.05 mmol) and **2a** (0.2 mmol) were heated in toluene (1.5 mL) at 110 °C for 12 h; then **3a** (0.1 mmol) and CPA* (0.0025 mmol) were added at room temperature (rt), and the reaction was allowed to proceed for 12 h. The dr value for

**4a** was >20:1. [b]Isolated yields are reported. [c]The ee values were determined by high-performance liquid chromatography with a chiral stationary phase. [d]Oxidative aromatization conditions: **4a** (0.05 mmol, 99% ee) and 1,2-dichloro-4,5-dicyano-benzoquinone (DDQ, 0.15 mmol) in solvent (2 mL) were allowed to react at rt for 3 h. [e]Conversion percentage (cp) = $ee_{4a}/ee_{5a} \times 100\%$.

chiral quinohelicenes with excellent enantioselectivities (Fig. 1D). Notably, the obtained quinohelicenes have rich optical properties.

## Results and discussion

The energetic barrier for interconversion between the two enantiomers of helically chiral compounds is known to depend strongly on the number of *ortho*-fused benzene rings, and helicenes with more than

five rings are usually stable. Therefore, to probe the feasibility of our strategy, we calculated the activation energy for racemization of the target quinohelicene, 1-(3-indol)-quino[5]helicene. To our delight, the calculated energy barrier was 36.7 kcal/mol (Fig. 2), which is 11.9 kcal/mol higher than the barrier for racemization of the corresponding hydrogen-substituted quino[5]helicene (Supplementary Fig. 4). This result indicated that the product we planned to synthesize would be

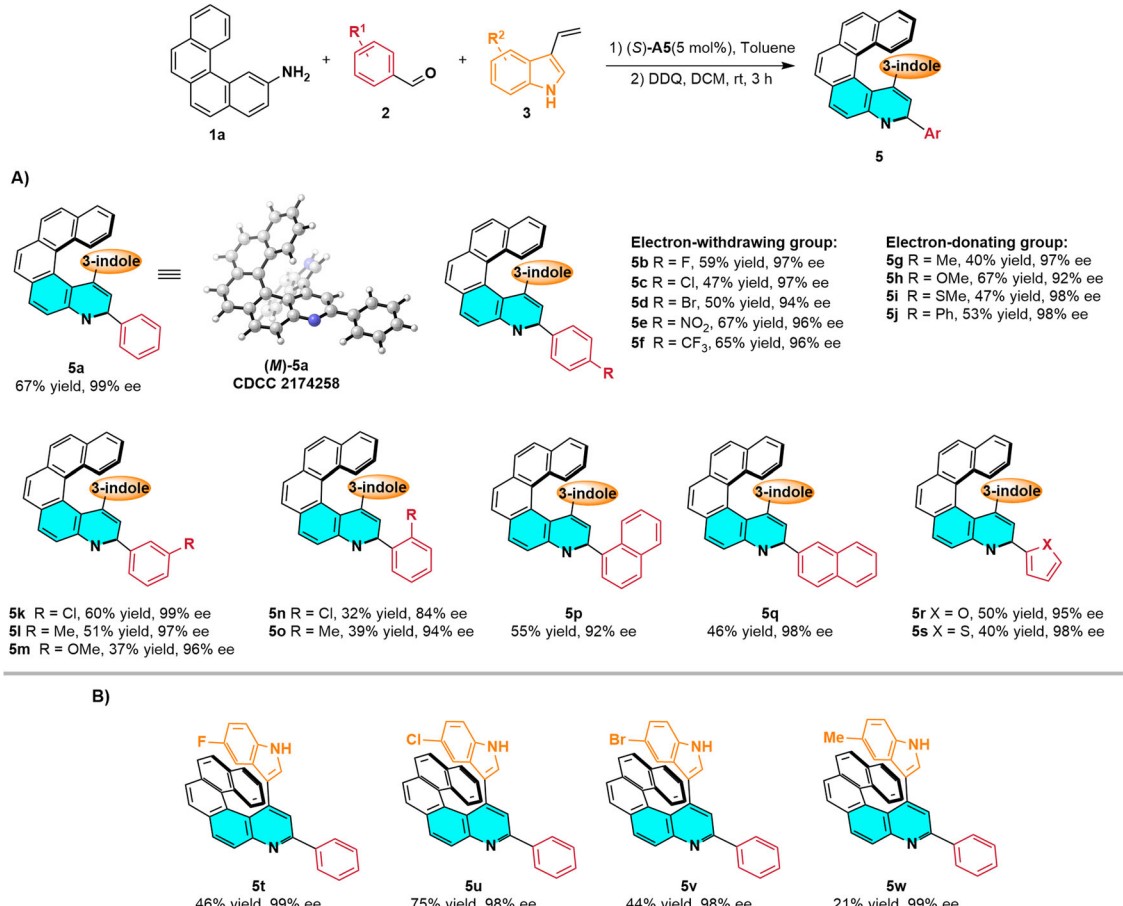

**Fig. 4 | Scope of the reaction.** Scope with respect to (**A**) aromatic aldehyde **2** and (**B**) indole **3**. Reaction conditions: (1) **1a** (0.1 mmol) and **2** (0.4 mmol) in toluene (2 mL) were heated at 110 °C for 12 h. After the mixture was cooled to room temperature, **3** (0.2 mmol) and (*S*)-**A5** (0.005 mmol) were added, and the reaction was allowed to proceed for 12 h. Silica gel column chromatography gave **4**. (2) Reaction of **4** and DDQ (0.3 mmol) in DCM (3 mL) at room temperature for 3 h afforded **5**.

stable at room temperature. With this information in hand, we designed and synthesized benzo[*c*]phenanthren-2-amine **1a** from inexpensive, readily available starting materials[36,65,66].

Next, we optimized the conditions for the Povarov reaction of **1a**, benzaldehyde (**2a**), and 3-vinyl-1*H*-indole (**3a**) as model substrates (Fig. 3A). Initially, the reaction was performed in toluene at 110 °C for 12 h in the presence of 5 mol % (*S*)-**A1** as the catalyst. Under these conditions, desired tetrahydroquinoline **4a** was obtained in 36% yield with 92% ee (entry 1). Subsequently, desired quinohelicene **5a** could be obtained in 87% yield with 90% ee by aromatization of **4a**. Inspired by this result, we assessed phosphoric acids (*S*)-**A2** to (*S*)-**C8** (entries 2–8) and found that (*S*)-**A5** was optimal, giving **4a** with 99% enantioselectivity (entry 5).

However, the ee for **5a** was highly unpredictable when the oxidative aromatization with DDQ was carried out in toluene (Supplementary Table 1). To address this issue, we screened various solvents and oxidants for the aromatization step (Fig. 3B and Supplementary Table 2). We were pleased to find that when dichloromethane (DCM) was used as the solvent and oxidize with DDQ, **5a** could be obtained in 91% yield with 99% ee with a 100% conversion percentage (Fig. 3B). To sum up, we obtained the optimal results by carrying out the enantioselective Povarov reaction in toluene to generate **4a** and then oxidizing **4a** with DDQ in DCM to generate **5a**.

With optimized conditions in hand, we explored the substrate scope of this reaction. Firstly, various aromatic aldehydes **2** were tested (Fig. 4A). *para*-Substituted benzaldehydes **2b**–**2j** gave corresponding quinohelicene products **5b**–**5j** in moderate yields (40–67%)

with very good enantioselectivities (92–98% ee) regardless of whether the *para* substituent was electron withdrawing (F, Cl, Br, NO₂, CF₃) or electron donating (Me, OMe, SMe, Ph). We also examined the outcomes of reactions of *meta*- and *ortho*-substituted benzaldehydes. These experiments revealed that compounds with electron-donating or electron-withdrawing substituents were well-tolerated, delivering products **5k**–**5o** with good to excellent enantioselectivities (84–99% ee). Naphthaldehydes were also investigated and found to afford quinohelicenes **5p** and **5q** with 92% ee and 98% ee, respectively. Furthermore, furfural and 2-thenaldehyde were compatible with the standard reaction conditions and delivered **5r** and **5s** with 95% ee and 98% ee, respectively.

To further explore the substrate scope, we evaluated reactions of **1a** and benzaldehyde (**2a**) with various indoles (Fig. 4B). We found that indoles with various substituents at C-5 were suitable for this Povarov/oxidation strategy, giving corresponding quinohelicenes **5t**–**5w** with 98–99% ee.

We next investigated reactions of various aromatic amine substrates **1**. Specifically, benzo[*c*]phenanthren-2-amines **1b**–**1i** were designed and synthesized and allowed to react with **2a** and **3a** (Fig. 5A). Gratifyingly, all the tested amines worked well under the optimized conditions, giving the desired products in 45–62% yields with 86–99% ee.

Further exploration of the substrate scope was focused on the helicene skeleton (Fig. 5B). Chromene-containing quinohelicenes **5aj**–**5al** were obtained from chromene-containing naphthalen-2-amines **1j**–**1l** in 46–81% yields with 87–97% ee. Furthermore,

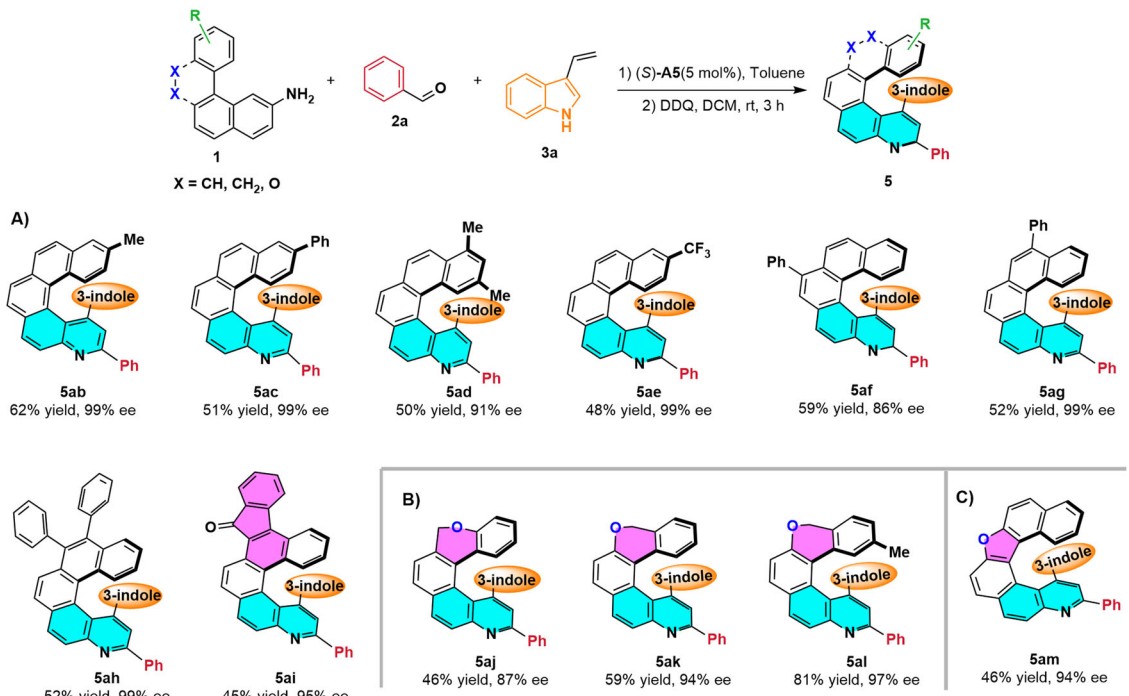

**Fig. 5 | Scope of the reaction.** Scope with respect to aromatic amines (**A**), (**B**), (**C**). Reaction conditions: (1) **1** (0.1 mmol) and **2a** (0.4 mmol) in toluene (2 mL) were heated at 110 °C for 12 h. After the mixture was cooled to room temperature, **3a** (0.2 mmol) and (*S*)-**A5** (0.005 mmol) were added, and the reaction was allowed to proceed for 12 h. Silica gel column chromatography gave **4**. (2) Reaction of **4** and DDQ (0.3 mmol) and DCM (3 mL) at room temperature for 3 h afforded **5**.

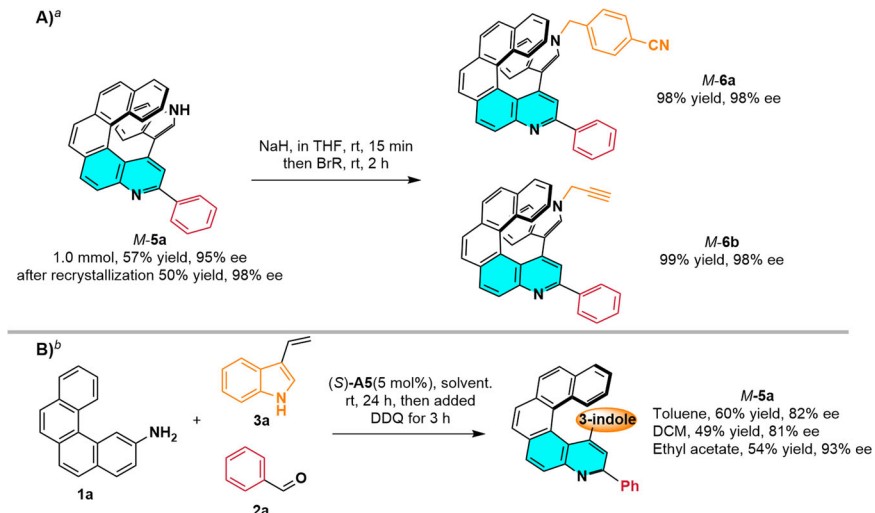

**Fig. 6 | Synthetic application.** Experiments about (**A**) Large-scale reaction and synthetic transformation, and (**B**) One-pot reaction. [a]The large-scale reaction was conducted under the optimized reaction conditions. [b]The one-pot reaction was conducted with **1a** (0.1 mmol), **2a** (0.4 mmol), **3a**, and (*S*)-**A5** (0.005 mmol) in solvent (2 mL) at rt for 24 h; then DDQ (0.5 mmol) was added, and the reaction was allowed to continue for another 3 h.

dinaphtho[2,1-*b*:1′,2′-d]furan-2-amine **1 m**, which has five rings, was also subjected to the reaction conditions and was found to give quinohelicene product **5am** in 46% yield with 94% ee. However, 7H-dibenzo[*c,g*]carbazol-2-amine (**1n**) failed to provide the corresponding aza-[6]quinohelicene under the standard conditions. The absolute configurations of **5a** and **5al** were assigned to be *M* on the basis of X-ray diffraction analysis, and the configurations of the other products were assigned by analogy (Supplementary Fig. 1 and Fig. 2).

To confirm that our strategy was useful for preparative-scale synthesis, we carried out a 1.0 mmol scale reaction of **1a, 2a**, and **3a** (Fig. 6A) and obtained **5a** in 57% yield with 95% ee (compare with 99%

ee for the small-scale reaction). To illustrate the synthetic utility of the method, we conducted some transformations of **5a** (Fig. 6A). Specifically, alkynyl- and cyano-quino[5]helicenes **6b** and **6a**, respectively, were obtained in nearly quantitative yields with no loss in optical purity. These chiral products can undergo click reactions and have potential applications in biometrics. To further explore the practicability of the method, we carried out this three-step transformation in one pot (Fig. 6B). We found that target chiral quinohelicene **5a** could be obtained with satisfactory enantioselectivity in toluene, DCM, or ethyl acetate. To our delight, the reaction in ethyl acetate gave **5a** in 54% yield with 93% ee.

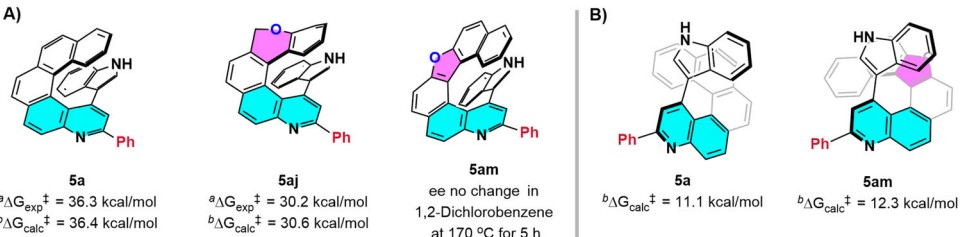

**Fig. 7 | Racemization experiments.** Barriers to racemization of (**A**) helical chirality and (**B**) axial chirality. [a]Experimental relative Gibbs free energy (kcal/mol). [b]Relative Gibbs free energies were calculated at the SMD(Toluene)/M06-2X-D3/def2-TZVPP// B3LYP D3(BJ)/def −2-SVP level of theory.

| [a]Compound | [b]$\lambda_{abs,\ max}$(nm) | [c]$\lambda_{em}/\tau$ (ns) ($\lambda_{ex}^1/\lambda_{ex}^2$ (nm)) | [d]$\phi_F$(%) ($\lambda_{ex}$(nm)) | [e]$g_{abs} = \Delta\varepsilon/\varepsilon$ ($\lambda_{abs}$(nm)) | [e]$g_{lum}$ ($\lambda_{em}/\lambda_{ex}$(nm)) |
|---|---|---|---|---|---|
| *M*-**5a** | 267, 325, 420 | 440/6.35, 450/6.36 (325/300) | 13.9(325) | 0.0016(305), -0.0056(388) | 0.0038(457/325) |
| *M*-**5ab** | 269, 325, 420 | 438/5.81, 457/6.06 (325/300) | 10.3(325) | 0.0020(310), -0.0072(390) | 0.0030(459/325) |
| *M*-**5ac** | 266, 330, 425 | 440/7.19, 461/7.29 (330/300) | 14.8(330) | 0.0008(315), -0.0083(393) | 0.0051(465/ 330) |
| *M*-**5ae** | 266, 315, 420 | 460/7.41 (315/300) | 9.8(315) | -0.0026(363), 0.0081(388) | 0.0024(468/315) |
| *M*-**5ah** | 265, 340, 424 | 436/6.84, 458/6.91 (340/300) | 26.0(340) | 0.0024(301), -0.0065(392) | 0.0015(460/340) |
| *M*-**5al** | 216, 320, 418 | 458/2.55 (320/300) | 11.3(320) | -0.0017(260), 0.0045(370) | 0.0030(465/320) |
| *M*-**5am** | 257, 313, 412 | 436/3.42 (313/300) | 25.7(313) | -0.0036(275), 0.0048(379) | 0.0013(477/313) |

**Fig. 8 | Photophysical properties of selected quinohelicenes. A** Summary of optical characterization data. **B** Solvent effect of **5a**. **C** Acid effect of **5a**. **D** Circular dichroic absorption (CD) of *M/P*-**5a** and *M/P*-**5al**. **E** Circular polarization luminescence (CPL) of *M/P*-**5a** and *M/P*-**5al**. [a]Measured at a concentration of $1 \times 10^{-5}$ M in DCM. [b]Maximum UV absorption. [c]Maximum fluorescence wavelength ($\lambda_{ex}^1$) and fluorescence lifetime ($\lambda_{ex}^2$). [d]Absolute fluorescence quantum efficiency for absorbance of < 0.1. [e]Measured for compounds with 99% ee. $g_{abs}$ = Absorption dissymmetry factors; $g_{lum}$ = luminescence dissymmetry factors.

To investigate the stability of the chiral quinohelicenes, we carried out some racemization experiments (Fig. 7A). We found that the racemization barriers for **5a** and **5aj** were 36.3 and 30.2 kcal/mol, respectively, and these values agreed well with the calculated values of 36.4 and 30.6 kcal/mol. The 5.8 kcal/mol difference between **5a** and **5aj** may have been due to the presence of the sp$^3$ carbon in **5aj**, which made the molecule more flexible. We also found that quino[6]helicene **5am** exhibited good configurational stability, and no obvious racemization was observed even after 5 h at 170 °C in 1,2-dichlorobenzene. Additionally, as shown in Fig. 7B, we also explored the stability of the axial chirality of **5a** and **5am** by DFT calculations. The comparison of the rotation barriers of **5a** (11.1 kcal/mol) and **5am** (12.3 kcal/mol) suggested the axial chirality of these compounds is extremely unstable at room temperature.

To demonstrate the potential applications of the synthesized quinohelicenes, we investigated the photophysical properties of selected molecules (Fig. 8A). Their fluorescence emission half-lives were determined in DCM, and oxygen-containing compounds **5al** and **5am** were found to have shorter half-lives than the compounds without oxygen. Notably, the quinohelicenes showed quantum yields ($\Phi_{FL}$) up to 26.0%. We also studied the effect of solvent on fluorescence and found that fluorescence gradually increased with increasing solvent polarity and showed a red shift (Fig. 8B). Moreover, the quantum yield of **5a** markedly increased with increasing solvent polarity, from 13.9% in DCM to 43.4% in DMSO (Supplementary Fig. 15). Because quinohelicenes exhibit high proton affinities, we studied the optical properties of **5a·H$^+$** in solutions containing 0–50 eq of trifluoroacetic acid. As the amount of acid was increased, the absorption at 240–390 nm gradually weakened; in contrast, that at 400–500 nm increased, and the fluorescence emission peak was markedly red shifted (Fig. 8C).

Finally, the chiroptical properties of *M*-**5a**, *M*-**5ab**, *M*-**5ac**, *M*-**5ae**, *M*-**5ah**, *M*-**5al**, and *M*-**5am**, were preliminarily evaluated (Fig. 8A). The circular dichroism spectra of these quinohelicenes displayed several signals at 230–400 nm and had fine absorption dissymmetry factors ($g_{abs}$); the maximum values for *M*-**5ab**, *M*-**5ac**, and *M*-**5ae** reached −0.0072, 0.0083, and 0.0081 at absorption wavelengths of 390, 393, and 388 nm, respectively. The circular dichroism spectrum of enantiomer *P*-**5a** was the mirror image of the spectrum of *M*-**5a**, and the spectrum of *P*-**5al** was the mirror image of that of *M*-**5al** (Fig. 8D). Additionally, the simulated CD spectra of the *M*-**5a** enantiomer have a good agreement with experimentally recorded CD spectra of *M*-**5a**. The circular polarized luminescence spectra of *M*-**5a**, *M*-**5ab**, *M*-**5ac**, and *M*-**5al** showed maximum luminescence dissymmetry factors ($g_{lum}$) of 0.0038, 0.0030, 0.0051, and 0.0030 at emission wavelengths of 457, 459, 465, and 465 nm, respectively. Moreover, *M*- and *P*-**5a** and *M*- and *P*-**5al** exhibited mirror-image circular polarized luminescence emissions (Fig. 8E).

In summary, we have developed a mild, reliable method for multicomponent organocatalytic enantioselective Povarov/aromatization reactions of benzo[*c*]phenanthren-2-amines, aromatic aldehydes, and 3-vinyl-1*H*-indoles to prepare highly functionalized quinohelicenes. The wide substrate scope, good yields, and high enantioselectivities make this a promising method for preparing chiral helicenes, which are widely used for applications in materials science, molecular recognition, and asymmetric catalysis. Moreover, the synthesized quinohelicenes showed enrich photophysical properties, including solvent and acid effects on UV and fluorescence, CD and CPL propeties.

## Data availability
The authors declare that the data supporting the findings of this study are available within the article and its Supplementary Information files. Additional data are available from the author upon request. Crystallographic data for the structures reported in this article have been deposited at the Cambridge Crystallographic Data Centre, under deposition numbers CCDC 2174258 (**5a**) and 2212769 (**5al**).

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

## Acknowledgements

The authors are grateful to the National Natural Science Foundation of China (Grant Nos. 22193011, 21971120, and 21933008) and National Science & Technology Fundamental Resource Investigation Program of China (No. 2018FY201200) for financial support. We are grateful to the support of the Supercomputing Center of Nankai University NKSC. We thank Shan Liu (Nankai University) for her contribution in measuring the redox potential and thank Qiu-Hong Huang (Nankai University) for her advice.

## Author contributions

C.L. conceived and designed the study, performed experiments, and wrote the manuscript; Y.-B.S. and Z.R. performed the theoretical calculations; X.G. synthesized some substrates; C.G. and M.L. performed the circular polarized luminescence experiments; X.L. conceived and designed the study and wrote the manuscript.

## Competing interests

The authors declare no competing interests.
