## [Peer Review File · Nature Communications]

Enantioselective Synthesis of Chiral Quinohelicenes through Sequential Organocatalyzed Povarov Reaction and Oxidative AromatizationReviewers' Comments:

Reviewer #1:

Remarks to the Author:

The manuscript submitted by Li and co-workers deals with the enantioselective organocatalyzed synthesis of heterohelicenes via a chiral phosphoric acid catalyzed three-component Povarov reaction between benzophenanthro-2-amines, aromatic aldehydes and 3-vinylindoles. This reaction affords a centrally chiral tetrahydroquinoline which are then oxidized with DDQ to give the desired aza-helicene thanks to a central-to-helical chirality conversion event. This consecutive two-steps methodology has been applied to various substrates and 34 examples of quino-helicenes have been obtained efficiently (21-81% yield, 84-99% ee). A few post-functionalizations have been realized and the physical (enantiomerization barrier) photophysical (fluorescence) and chiroptical (CD and CPL spectrum) properties of these molecules have been investigated.

In conclusion, this is a nice piece of well-executed work which meets the requirements of novelty and originality necessary for publication in Nature Communications. Only few organocatalysed reactions allow to stereoselectively construct helicenes. This methodology is very attractive and complementary to existing enantioselective methodologies in particular in the fact that the pyridine ring is in an external position of the helicene structure. In addition these new helicenes are easily accessed in a one pot procedure. Consequently I give my support for publication in Nature Communications after addressing these major points.

- 1) Apparently, product 4a has been isolated as one single diastereomer. This is not clearly indicated and the relative and absolute stereochemistry is not given.
 - 2) The authors indicate a central-to-helical chirality conversion during the oxidation step. This may be true, but in fact, this would require that the helicity is not present on the tetrahydroquinoline intermediate 4a. Given the high steric hindrance introduced during the Povarov reaction, the molecule may already adopt an helical shape. This could be checked by X-Ray diffraction analysis of 4a or another intermediate 4. If it is the case, this would not be a central-to-helical chirality conversion but rather these two stereogenic elements would be introduced simultaneously in the first step, and the oxidation step would destroy the central chirality, while keeping the helicity with almost full retention.
 - 3) It seems that this family of heterohelicenes may display additional axial chirality, due to slow rotation of the indole ring, in particular compound 5am with an extra ring. Has this been checked by theoretical studies or low/high temperature NMR analysis? If this is not the case, the construction of another heterohelicenes starting from 2-substituted-3-vinylindole would certainly bring the steric bulk necessary to prevent rotation around this bond.
 - 4) No comment was given on the oxidation step, its optimization, in particular in the effect of alternative oxidation reagents. As it has been observed in the case of the central-to-axial chirality conversion, different oxidation reagents may lead to opposite atropisomers. This would be very interesting and unprecedented to observe such behavior in the case of central-to-helical conversion.
- 4) Minor corrections :
- "another effective approach for the synthesis of chiral helicenes..." should be replaced by "another effective approach for the synthesis of enantioenriched helicenes..."
 - "To date, only two examples of..." should be replaced by "To date, only two examples of..."
 - "Bronsted" should be replaced by "Brønsted"

Reviewer #2:

Remarks to the Author:

In this manuscript, Li and coworkers described a mild and multicomponent organocatalytic enantioselective Povarov/aromatization reaction of benzo[c]phenanthren-2-amines, benzaldehydes and 3-vinyl-1H-indoles, in which provide a reliable tool for the preparation of highly functionalized quino-helicenes in high to excellent enantioselectivities with a CPA catalyst followed by an oxidative central-to-helical chirality conversion process. Besides, the synthesized quino-helicenes showed enriched photophysical properties, including solvent and acid effects on UV and fluorescence, circular dichroism

absorption (CD) and circular polarization luminescence (CPL) properties. However, central-to-helical conversion process in asymmetric catalysis has a few examples. Meanwhile, the catalytic enantioselective synthesis of helicenes was well explored by whether transition metal catalyzed or organocatalyzed (limited examples). As such, this work may be suitable for publication in Nature Communications after major revisions:

Items that should be addressed:

- a) The paper contains some grammatical and stylistic errors that should be carefully re-edited.
- b) Have the authors tried the substrate with other benzoheterocyclic moiety rather than indole moiety in this reaction?
- c) In Figure 4C, the authors used dinaphtho[2,1-b:1',2'-d]furan-2-amine to obtain the corresponding oxy-[6]quinohelicene product 5am. If 7H-dibenzo[c,g]carbazol-2-amine is used, can the corresponding aza-[6]quinohelicene product be obtained?
- d) In the Povarov reaction, the authors used only one type of electron-rich olefin, 3-vinyl-1H-indole, what about other types of electron-rich olefins? Can intracyclic olefins such as norbornene be used?
- e) In addition to a stable helicene in the products reported by the authors, there may be a relatively unstable axial chirality (indole moiety), and the authors can further explore the stability of the possible axial chirality by calculations or experiments and compare the stability of axial chirality in [5]helicene and [6]helicene (5am).
- f) The fluorescence lifetime of the resulting product should also be given.
- g) Characterize the electrochemical properties of the products, such as testing CV voltammogram, etc.?
- h) In NMR data spectrum of SI, there are noticeable amounts of impurities or grease in proton NMR.

Reviewer #3:

Remarks to the Author:

Dr. Li and co-authors described a straightforward method from amino[4]helicene to reach a rich array of enantioenriched quino[5]helicenes with high enantiomerization barrier using Povarov reaction. In addition, the reactions are in good yield and high enantioselectivity, and various substrates including different helicene skeletons, aldehydes and vinyl-indole derivatives were investigated. Finally, the author demonstrated the potential application value by studying the optical and chiroptical properties with solvent effect and acid sensing. The supporting information is also well prepared. However, my concern is about the novelty and significance of the results. In my opinion, the present results do not represent an important advance within the field of helicene synthesis. The authors mentioned that this reaction avoids the use of transition metal but without any emphasizing for the importance of organocatalysts in the introduction to support this work (only the list of previous works). For me, I don't think this point will be important for synthesizing chiral helicenes as there is not much economic and pharmaceutical concerns in helicene synthesis. More importantly, the detailed mechanism is lacking. It will be much interesting and enlightening to investigate how the chiral catalyst interact with helicene's helical skeleton to reach the high enantioselectivity. Besides, I also concern the scope of the reaction. Although the authors tried various helicene skeletons, aldehydes and vinyl-indoles for the Povarov reaction, all the products contain 3-indole groups. I don't get the importance to introduce this indole group, because from my point, for typical Povarov reactions, many functional groups such as various alkyl or aryl groups even alkoxy groups could work, and trying with various other functional groups might be more favored, for studying the enantiomerization process as well as the optical and chiroptical properties. Therefore, I do not think that the manuscript merits publication in Nature Communications, and I suggest the author to submit this work in a more specific journal.

What is more, there are several problems should also be addressed:

1. Although the author did not discuss the detailed reaction mechanism, I suggest the author at least discuss the regioselectivity of this Povarov reaction. For benzo[c]phenanthren-2-amine 1a, it may have two candidate positions for the Povarov reaction (two ortho-positions near the amine group),

while in this manuscript, it reports the products that the reaction only took place on the inner position, which seems to be sterically unfavored. The author should discuss why the Povarov reaction chose this sterically unfavored position. Similarly, for the ethylenes, it may also have two orientations during the Povarov reaction (the 3-indole group could be in the inner position as in your work but with steric effect from the other terminal ring of helicene, or in the outer position), the authors should also discuss about it.

2. The author mentioned that "The absolute configuration of 5a and 5al were both assigned as M based on an X-ray diffraction analysis". I checked the SI, and the Flack parameters are 0.10(8) and -0.12(10) which are not precise enough for determining the absolute configurations. This is due to the lack of heavy atoms in the crystals. I suggest to use TD-DFT calculations for determining the absolute configurations as you already have both enantiomers and their ECD. You just need calculate the ECD of one enantiomer and make the comparison.

3. For the theoretical calculation part, the transition state should also be verified by intrinsic reaction coordinate (IRC) to ensure that the TS rather connects with the two local minima since in your case the enantiomerization might be more complicated due to the additional indole group. Besides, I strongly encourage the authors to add Grimme's D3 dispersion for the geometry optimization, because in your case you have obvious pi-pi interaction in your structures and only use B3LYP is obviously not sufficient (you need to use B3LYP-D3 or B3LYP-D3(BJ) for calculations).

Thank you for giving us an opportunity to revise our paper. We really appreciate the reviewers' valuable comments, which are helpful to improve the quality of our study. According to the comments, we have revised our paper as follows:

Reviewer #1 (Remarks to the Author):

1) Apparently, product **4a** has been isolated as one single diastereomer. This is not clearly indicated and the relative and absolute stereochemistry is not given.

*We thank for this reviewer's valuable suggestion. Following the reviewer's comments, we attempted to cultivate single crystals of **4a** to determine the relative and absolute stereochemistry. Unfortunately, we have tried many methods without success. At the same time, we also attempted to cultivate single crystals of other **4** (**4c**, **4d**, **4v**), and didn't get a satisfactory result either. However, based on the research on the configuration and mechanism of similar compounds in the literature (see the Reply to question 2), combined with the absolute configuration of product **5a**, we speculate that the configuration of **4a** should be M, S, R. For details, please see the revised supplementary information.*

2) The authors indicate a central-to-helical chirality conversion during the oxidation step. This may be true, but in fact, this would require that the helicity is not present on the tetrahydroquinoline intermediate **4a**. Given the high steric hindrance introduced during the Povarov reaction, the molecule may already adopt an helical shape. This could be checked by X-Ray diffraction analysis of **4a** or another intermediate **4**. If it is the case, this would not be a central-to-helical chirality conversion but rather these two stereogenic elements would be introduced simultaneously in the first step, and the oxidation step would destroy the central chirality, while keeping the helicity with almost full retention.

*We thank for this reviewer's valuable suggestion. Unfortunately, we can't get the single crystal of **4a** or others intermediate **4** (**4c**, **4d**, **4v**). But according to reviews of enantioselective Povarov reactions reported by Benaglia et al. (Eur. J. Org. Chem. 2022, e202101171) and Menéndez et al. (Org. Biomol. Chem., 2022, 20, 1550–1581), and works reported by Ricci et al. (Chem. Commun., 2010, 46, 327–329), Shi et al. (J. Org. Chem. 2016, 81, 185–192) and Corti et al. (Chem. Eur. J. 2019, 25, 15694–15701), and absolute configuration of M-**5a**, we can speculate that the absolute configuration of **4a** should be M, S, R. Additionally, the helical chiral racemization barriers for **4a** is 29.4 kcal/mol by DFT calculations. As the reviewer said, helical and central chirality would be simultaneous controlled in the first step, and then oxidizing M, S, R-**4a** with DDQ to generate M-**5a** by destroy the central chirality but keeping the helicity. The similar observation was reported by Bonne and Rodriguez et al. (J. Am. Chem. Soc. 2020, 142, 16199–16204). Therefore, we have made revisions to this expression of "central-to-helical chirality conversion". For details, please see the revised manuscript and supplementary information.*

3) It seems that this family of heterohelicenes may display additional axial chirality, due to slow rotation of the indole ring, in particular compound **5am** with an extra ring. Has this been checked by theoretical studies or low/high temperature NMR analysis? If this is not the case, the construction

of another heterohelicenes starting from 2-substituted-3-vinylindole would certainly bring the steric bulk necessary to prevent rotation around this bond.

*We thank for this reviewer's valuable suggestion. During the preparation of this revised manuscript, we have explored the stability of the axial chirality of **5a** and **5am** by DFT calculations. The comparison of the rotation barrier of **5a** (11.1 kcal/mol) and **5am** (12.3 kcal/mol) suggested the axial chirality of these compounds is unstable. We have tried to construct more stable axial chirality quinoelicenes starting from 2-methyl-3-vinyl-1H-indole or 2-phenyl-3-vinyl-1H-indole, but there is no corresponding product be formed. For details, please see the revised manuscript and supplementary information.*

4) No comment were given on the oxidation step, its optimization, in particular in the effect of alternative oxidation reagents. As it has been observed in the case of the central-to-axial chirality conversion, different oxidation reagents may lead to opposite atropisomers. This would be very interesting and unprecedented to observe such behavior in the case of central-to-helical conversion.

We thank for this reviewer's valuable suggestion. During the preparation of this revised manuscript, we carried out the solvent screening for the oxidation step in Fig.3B. And according to your comments, we have replaced DDQ by others oxidation reagents. For details, please see the Table S2 in the revised supplementary information.

5) Minor corrections :

- "another effective approach for the synthesis of chiral helicenes..." should be replaced by "another effective approach for the synthesis of enantioenriched helicenes..."
- "To data, only two examples of..." should be replaced by "To date, only two examples of..."
- "Bronsted" should be replaced by "Brønsted"

We thank for this reviewer's carefulness. And we have corrected these errors. Please see the revised manuscript.

Reviewer #2 (Remarks to the Author):

a) The paper contains some grammatical and stylistic errors that should be carefully re-edited.

We thank for this reviewer's carefulness. And this article was carefully reviewed and corrected for grammatical and stylistic errors. Please see the revised manuscript.

b) Have the authors tried the substrate with other benzoheterocyclic moiety rather than indole moiety in this reaction?

We thank for this reviewer's valuable suggestion. During the preparation of this revised manuscript, 3-vinylbenzofuran and 3-vinylbenzo[b]thiophene were tried in this reaction, but no corresponding products have been obtained.

c) In Figure 4C, the authors used dinaphtho[2,1-b:1',2'-d]furan-2-amine to obtain the corresponding oxy-[6]quinohelicene product 5am. If 7H-dibenzo[*c,g*]carbazol-2-amine is used, can the corresponding aza-[6]quinohelicene product be obtained?

*We thank for this reviewer's valuable suggestion. During the preparation of this revised manuscript, we attempted to synthesize the substrate suggested by the reviewer. Through a lot of effort, 7H-dibenzo[*c,g*]carbazol-2-amine was obtained through the synthesis route shown in the following Figure. Unfortunately, under standard conditions, 7H-dibenzo[*c,g*]carbazol-2-amine did not undergo the expected reaction to obtain the corresponding aza-[6]quinohelicene product.*

7H-dibenzo[*c,g*]carbazol-2-amine:

¹H NMR (400 MHz, DMSO-*d*₆) δ 12.03 (s, 1H), 9.10 (d, *J* = 8.5 Hz, 1H), 8.12 (s, 1H), 8.05 (d, *J* = 8.0 Hz, 1H), 7.85 (d, *J* = 8.7 Hz, 1H), 7.78 – 7.72 (m, 2H), 7.69 – 7.61 (m, 2H), 7.48 (t, *J* = 7.4 Hz, 1H), 7.40 (d, *J* = 8.5 Hz, 1H), 6.89 (dd, *J* = 8.6, 2.1 Hz, 1H), 5.57 (s, 2H).

¹³C NMR (101 MHz, DMSO-*d*₆) δ 147.32, 137.79, 136.54, 131.34, 130.67, 129.75, 129.49, 129.10, 127.04, 125.88, 125.59, 125.39, 122.87, 122.42, 117.14, 115.40, 114.55, 113.83, 108.73, 106.18.

d) In the Povarov reaction, the authors used only one type of electron-rich olefin, 3-vinyl-1H-indole, what about other types of electron-rich olefins? Can intracyclic olefins such as norbornene be used?

We thank for this reviewer's valuable suggestion. During the preparation of this revised manuscript, we tried other types of electron-rich olefins and the results shown below. When used 2-vinylphenol, corresponding quinohelicene product can be obtained in 47% yield with 3% ee. However, other olefins do not work.

2-(3-phenylphenanthro[4,3-f]quinolin-1-yl)phenyl 4-methylbenzenesulfonate

^1H NMR (400 MHz, $\text{DMSO-}d_6$) δ 8.41 – 8.34 (m, 2H), 8.30 (d, J = 8.7 Hz, 1H), 8.17 (d, J = 8.6 Hz, 1H), 8.08 (d, J = 8.1 Hz, 1H), 8.04 (d, J = 8.2 Hz, 1H), 7.80 (d, J = 8.6 Hz, 1H), 7.72 (d, J = 8.5 Hz, 1H), 7.67 – 7.52 (m, 5H), 7.33 (t, J = 7.4 Hz, 1H), 7.16 (t, J = 7.7 Hz, 1H), 6.98 – 6.87 (m, 2H), 6.82 – 6.69 (m, 4H), 6.41 (d, J = 8.2 Hz, 1H), 6.18 (d, J = 8.1 Hz, 2H), 1.68 (s, 3H).

^{13}C NMR (101 MHz, CDCl_3) δ 155.24, 148.22, 144.53, 144.05, 143.70, 138.57, 134.07, 132.03, 131.41, 131.02, 130.89, 130.37, 129.68, 129.56, 129.08, 128.94, 128.11, 127.85, 127.28, 127.11, 127.05, 126.91, 126.62, 125.68, 125.63, 125.51, 125.32, 123.97, 122.53, 119.22, 20.85.

e) In addition to a stable helicene in the products reported by the authors, there may be a relatively unstable axial chirality (indole moiety), and the authors can further explore the stability of the possible axial chirality by calculations or experiments and compare the stability of axial chirality in [5]helicene and [6]helicene (**5am**).

*We thank for this reviewer's valuable suggestion. we have explored the stability of the axial chirality of **5a** (11.1 kcal/mol) and **5am** (12.3 kcal/mol) by DFT calculations. For details, please see the revised manuscript and supplementary information.*

f) The fluorescence lifetime of the resulting product should also be given.

*We thank for this reviewer's carefulness. We have tested the fluorescence lifetime of some products (**5a**, **5ab**, **5ac**, **5ae**, **5ah**, **5al**, **5am**). **5al** and **5am** displayed shorter fluorescence lifetime than other products. Corresponding results were added in Fig.8A. For details, please see the revised manuscript.*

g) Characterize the electrochemical properties of the products, such as testing CV voltammogram, etc.?

*We thank for this reviewer's valuable suggestion. The CV voltammogram of some products (**5a**, **5aj**, **5am**) were tested. For details, please see supplementary information.*

h) In NMR data spectrum of SI, there are noticeable amounts of impurities or grease in proton NMR.

We thank for this reviewer's carefulness. Compounds (**5d**, **5m**, **5f**, **5w**, **5ac** and **5ag**) were re-purified. For details, please see the revised supplementary information.

Reviewer #3 (Remarks to the Author):

1. Although the author did not discuss the detailed reaction mechanism, I suggest the author at least discuss the regioselectivity of this Povarov reaction. For benzo[*c*]phenanthren-2-amine **1a**, it may have two candidate positions for the Povarov reaction (two ortho-positions near the amine group), while in this manuscript, it reports the products that the reaction only took place on the inner position, which seems to be sterically unfavored. The author should discuss why the Povarov reaction chose this sterically unfavored position. Similarly, for the ethylenes, it may also have two orientations during the Povarov reaction (the 3-indole group could be in the inner position as in your work but with steric effect from the other terminal ring of helicene, or in the outer position), the authors should also discuss about it.

We thank for this reviewer's valuable suggestion. We have calculated the regioselectivity of this Povarov reaction as show below. Obviously, **TS1** reaction mode is more advantageous than **TS1-b**. And the similar observation was reported by Ricci et al. (Chem. Commun., 2010, 46, 327–329), Shi et al. (J. Org. Chem. 2016, 81, 185–192) and Corti et al. (Chem. Eur. J. 2019, 25, 15694–15701). Additionally, this result can also be supported by reviews reported by Benaglia et al. (Eur. J. Org. Chem. 2022, e202101171) and Menéndez et al. (Org. Biomol. Chem., 2022, 20, 1550–1581). And to gain insights into the mechanism and the enantioselectivity, we performed computational studies at the SMD(Toluene)/M06-2X-D3/def2-TZVPP//B3LYP D3(BJ)/def-2-SVP level of theory. For details, please see the revised manuscript and supplementary information.

2. The author mentioned that “The absolute configuration of **5a** and **5al** were both assigned as M based on an X-ray diffraction analysis”. I checked the SI, and the Flack parameters are 0.10(8) and -0.12(10) which are not precise enough for determining the absolute configurations. This is due to the lack of heavy atoms in the crystals. I suggest to use TD-DFT calculations for determining the absolute configurations as you already have both enantiomers and their ECD. You just need calculate the ECD of one enantiomer and make the comparison.

We thank for this reviewer's valuable suggestion. We have calculated the CD of **M-5a**. As shown in the figure below, the CD spectrum of enantiomer **5a** measured experimentally has a good

consistency with ECD spectrum of *M*-**5a** simulated by calculation, and **5a** can be determined to be *M* configuration.

3. For the theoretical calculation part, the transition state should also be verified by intrinsic reaction coordinate (IRC) to ensure that the TS rather connects with the two local minima since in your case the enantiomerization might be more complicated due to the additional indole group. Besides, I strongly encourage the authors to add Grimme's D3 dispersion for the geometry optimization, because in your case you have obvious pi-pi interaction in your structures and only use B3LYP is obviously not sufficient (you need to use B3LYP-D3 or B3LYP-D3(BJ) for calculations).

We thank for this reviewer's valuable suggestion. We have verified the transition state by intrinsic reaction coordinate (IRC), and change the calculation level to SMD(Toluene)/M06-2X-D3/def2-TZVPP//B3LYP D3(BJ)/def -2-SVP. For details, please see the revised manuscript and supplementary information.

Moreover, other changes request by the editor was also conducted. Please see the revised manuscript.

Sincerely,

Xin Li

Reviewers' Comments:

Reviewer #1:

Remarks to the Author:

I appreciate the modifications and additional experiments done by the authors. This study represents significant progress and novelty to justify its publication in Nature Communications.

Reviewer #2:

Remarks to the Author:

In this response, Li and coworkers have perfectly addressed the queries raised by the reviewers, and therefore I consider the manuscript suitable for publication in Nature Communications.

Reviewer #3:

Remarks to the Author:

The authors addressed my concerns, and revised the manuscript as well as supporting information with detailed additional calculations on reaction mechanism and the enantiomerization process together with discussion, which appears much better compared with the first version. Therefore, I would like to recommend this work to be published in Nature Communications.